# Extent of Left Ventricular Mass Regression and Impact of Global Left Ventricular Afterload on Cardiac Events and Mortality after Aortic Valve Replacement

**DOI:** 10.3390/jcm11247482

**Published:** 2022-12-16

**Authors:** Jer-Shen Chen, Jih-Hsin Huang, Kuan-Ming Chiu, Chih-Yao Chiang

**Affiliations:** 1Department of Cardiovascular Surgery, Cardiovascular Center, Far Eastern Memorial Hospital, New Taipei City 220216, Taiwan; 2Department of Healthcare Administration, Asian Eastern University of Science and Technology, New Taipei City 220303, Taiwan; 3Department of Applied Cosmetology, LeeMing Institute of Technology, New Taipei City 243083, Taiwan; 4Department of Electrical Engineering, Yuan Ze University, Taoyuan 320315, Taiwan; 5Division of Cardiovascular Surgery, Department of Surgery, School of Medicine, National Defense Medical Center, Taipei 114201, Taiwan

**Keywords:** aortic stenosis, aortic valve replacement, patient-prosthesis mismatch, left ventricular mass regression, reverse remodel, relative wall thickness, stroke work loss, Z_VA_, MACCE

## Abstract

Patient-prosthesis mismatch (PPM) causes a high transvalvular pressure gradient and residual left ventricular (LV) hypertrophy, consequently influencing long-term results. This study aimed to find the relationships between hemodynamic parameters and LV mass regression and determine the risk predictors of major adverse cardiovascular and cerebral events (MACCE) after aortic valve replacement (AVR) for aortic stenosis. *Methods and Results:* Preoperative and postoperative Doppler echocardiography data were evaluated for 120 patients after AVR. The patients’ mean age was 67.7 years; 55% of the patients were male. Forty-four (37%) patients suffered from MACCE during a mean follow-up period of 3.6 ± 2 years. The following hemodynamic parameters at follow-up were associated with lower relative indexed LV mass (LVMI) regression: lower postoperative indexed effective orifice area, greater mean transvalvular pressure gradient (MPG), greater stroke work loss (SWL), and concentric or eccentric LV remodeling mode. The following hemodynamic parameters at follow-up were associated with a higher risk of MACCE: higher valvuloarterial impedance (Z_VA_), greater SWL, greater MPG, greater relative wall thickness, greater LVMI, and hypertrophic LV remodeling mode. Lower relative LVMI regression was associated with a higher risk of MACCE (hazard ratio, 1.01: 95% confidence interval, 1.003–1.03). The corresponding cutoff of relative LVMI regression was −14%. *Conclusions:* Changes in hemodynamic parameters were independently associated with relative LVMI regression. Impaired reverse remodeling and persistent residual LV hypertrophy were independent risk predictors of MACCE. An LVMI regression lower than 14% indicated higher MACCE. A postoperative Z_VA_ greater than 3.5 mmHg/mL/m^2^ was an independent risk predictor of cardiac events and mortality after AVR. Preventive strategies should be used at the time of the operation to avoid PPM.

## 1. Introduction

Degenerative aortic stenosis (AS) is the most prevalent heart valve disorder in developed countries [1]. Hypertrophic left ventricular (LV) remodeling is an adaptive response to the pressure overload caused by severe AS and has worse outcomes in LV hypertrophy [2]. LV hypertrophy is associated with an increase in cardiovascular adverse events and increased LV mass is a risk predictor of mortality [3,4,5,6]. The purpose of aortic valve replacement (AVR) is to unload the LV burden and normalize LV mass to enable reverse remodeling of the hypertrophic left ventricle [7,8].

The concept of patient-prosthesis mismatch (PPM) was first proposed by Rahimtoola in 1978 [9]. PPM is defined as an indexed effective orifice area (EOAI) relative to the body surface area of less than 0.85 cm^2^/m^2^. PPM has the hemodynamic consequence of an elevated transvalvular pressure gradient with normally functioning aortic valves [10]. PPM is related to lower LV mass regression and impaired reverse remodeling, and consequently residual LV hypertrophy, indicating a higher risk of adverse events and mortality after AVR [11,12]. In previous studies, the prevalence of moderate PPM following AVR was 20–70% and of severe PPM was 2–11% [13,14]. In most Asian patients, a small-sized valve will be implanted due to their short stature and smaller aortic annuli.

Doppler echocardiography is a noninvasive method of evaluating cardiac geometry and function. It is a conventional tool for preoperative grading of AS severity and postoperative assessment of AVR results. Hypertension is a common comorbidity in AS with negative LV remodeling. Stroke work loss (SWL) is the amount of LV mechanical energy dissipated as heat and kinetic energy because of LV outflow obstruction [15]. Valvuloarterial impedance (Z_VA_) is an indicator of the global LV afterload [16]. SWL and Z_VA_ can function as risk stratification markers to predict LV dysfunction and major adverse cardiac and cerebral events [17,18]. This study aimed to assess the associations between hemodynamic parameters and relative left ventricular mass index (LVMI) regression and the influence of risk predictors on major adverse cardiovascular and cerebral events (MACCE) after surgical AVR for pure AS.

## 2. Materials and Methods

### 2.1. Study Population and Design

This study recruited 175 patients who underwent AVR for severe AS between January 2015 and December 2020. One hundred and fifty-seven patients remained after excluding those with other conditions or those undergoing other combined operations except for coronary artery bypass graft surgery. The exclusion criteria were larger than moderate aortic or mitral regurgitation, mitral valve surgery, aortic root surgery, myectomy, ascending aorta pathology, and infective endocarditis. Of these patients, only 120 had tracked Doppler echocardiography data; therefore, the final study population included these patients who underwent AVR for pure AS. The patients’ data were retrieved from their electronic medical records and from subsequent clinic visits. The study procedures were performed according to the guidelines stipulated in the Declaration of Helsinki, and the study was approved by the Institutional Review Board of Far Eastern Memorial Hospital (approval number: 111256-E).

### 2.2. Definition of Comorbidity and Hospital Complications and Mortality

We considered patients with hypertension to be those who had a history of hypertension or were treated with antihypertensive drugs. Chronic renal insufficiency was defined as a serum creatinine level of >2.0 mg/dL. A previous stroke event was defined as having a history of a central neurologic deficit persisting for more than 72 h, diagnosed by a neurologist, and confirmed by brain computed tomography (CT) or magnetic resonance imaging. Diabetes was defined as a history of diabetes mellitus or the need for an of oral hypoglycemic agent or insulin treatment. Chronic obstructive pulmonary disease was defined as the need for pharmacologic therapy or preoperative respirometry for a diagnosis of moderate or severe obstructive pulmonary disease. Peripheral artery disease was defined as the symptom of intermittent claudication confirmed by CT angiography. Carotid stenosis was also confirmed by CT angiography. Hyperlipidemia was defined as a low-density lipoprotein cholesterol level of >100 mg/dL.

Postoperative renal failure was defined as creatinine >2 mg/dL increase in the baseline creatinine level in the absence of end-stage renal failure on dialysis. Respiratory failure was regarded as the need for respiratory support for more than 48 h. Low cardiac output syndrome was defined as the use of postoperative inotropic support for more than 24 h. In-hospital mortality was defined as death at any time before discharge from the hospital. The survivors were followed up at the outpatient department and were subject to a scheduled echocardiographic survey at our hospital.

### 2.3. Definition of Hemodynamic Parameters

All patients were examined using two-dimensional Doppler echocardiography 0 to 7 days preoperation and 6 months to 2 years postoperation. The assessment was performed by experienced echocardiographers using an ultrasound system (Philips iE33, Philips Medical Systems, Andover, MA, USA) equipped with 2.5- to 3.5-MHz transducers.

The geometric measurements including LV end-diastolic diameter (LVEDD), LV end-systolic diameter (LVESD), end-diastolic ventricular septal thickness, and end-diastolic LV posterior wall thickness (PWT) were calculated using M-mode echocardiography. LV end-diastolic volume (LVEDV) and end-systolic volume (LVESV) were calculated from the apical two-chamber and four-chamber views using a modification of Simpson’s method or Teichholz formula [19,20,21,22].

Left ventricular mass (LVM) was calculated using the corrected American Society of Echocardiography formula as follows [23]:(1)$$\mathrm{LVM}=0.8[1.04(({\mathrm{IVS}}_{d}+{\mathrm{LVID}}_{d}+{\mathrm{PWT}}_{d}{)}^{3}-{{\mathrm{LVID}}_{d}}^{3})]+0.6$$

The indexed LV mass was indexed to body surface area.

LV systolic function ejection fraction was calculated using the following formula:(2)$$\mathrm{LV}\;\mathrm{ejection}\;\mathrm{fraction}=\left[\left(\mathrm{LVEVd}-\mathrm{LVEVs}\right)÷\mathrm{LVEVd}\right]\times 100$$

The relative wall thickness (RWT) ratio was calculated using the following formula [4,5]:(3)RWT ratio=(2×PWTd) ÷ LVIDd

The peak and mean transvalvular flow velocities were measured using continuous-wave Doppler echocardiography and were calculated using the modified Bernoulli equation. LV outflow tract (LVOT) was measured in mid-systole from the parasternal long-axis view according to standard criteria.

Stroke volume was calculated as the product of LVOT area and the LVOT pulsed Doppler velocity–time integral. LV cardiac output was calculated as the product of the heart rate and stroke volume and the indexed stroke volume was calculated by indexed body surface area [21].

Effective Orifice Area (EOA) calculation:

The preoperative aortic valve area was calculated using the continuity equation (stroke volume divided by valve flow velocity–time integral as determined by continuous-wave Doppler).
(4)$$\mathrm{Effective}\;\mathrm{orifice}\;\mathrm{area}=({\mathrm{Area}}_{\mathrm{LVOT}}\times {\mathrm{VTI}}_{\mathrm{LVOT}})\;÷\;{\mathrm{VTI}}_{\mathrm{AV}}$$
(5)Indexed Effective orifice area=(Effective orifice area) ÷ body surface area

The LV SWL was given as a percentage and calculated using the following equation [15]:(6)$$\mathrm{SWL}\;=\left[\mathrm{mean}\;\mathrm{pressure}\;\mathrm{gradient}\;÷\left(\mathrm{mean}\;\mathrm{pressure}\;\mathrm{gradient}+\mathrm{systolic}\;\mathrm{arterial}\;\mathrm{pressure}\right)\right]\times 100$$

The regression ratio of LV mass was defined as
(7)$$\mathrm{Regression}\;\mathrm{ratio}=\left[\left(\mathrm{postoperative}\;\mathrm{LV}\;\mathrm{mass}\;\mathrm{index}-\mathrm{preoperative}\;\mathrm{LV}\;\mathrm{mass}\;\mathrm{index}\right)÷\mathrm{preoperative}\;\mathrm{LV}\;\mathrm{mass}\;\mathrm{index}\right.\left.\;\right]\times 100$$
and then indexed to body surface area for relative indexed LV mass regression.

The global LV afterload valvuloarterial impedance (Z_VA_) was the sum of a double load, the valvular load caused by AS, and the arterial load determined by a decrease in systemic arterial compliance. Accordingly, the ZVA was calculated as follows [20]:(8)$$Z_{\mathrm{VA}}=(\mathrm{systolic}\;\mathrm{arterial}\;\mathrm{pressure}+\mathrm{mean}\;\mathrm{transvalvular}\;\mathrm{pressure}\;\mathrm{gradient})\;÷\;\mathrm{stroke}\;\mathrm{volume}\;\mathrm{index}\;$$
Alteration of hemodynamic parameter=(postoperative value−preoperative value)

Hemodynamic parameters were as follows: EOAI, MPG, SWL, Z_VA_, LVMI, RWT

### 2.4. Definition of Patient-Prosthesis Mismatch

The postoperative effective orifice area (EOA) was adapted from the EOA projected in a previous study [24] and then indexed to body surface area to assess the severity of PPM [8,25,26]. PPM was severe if EOAI < 0.65 cm^2^/m^2^, moderate if EOAI ≥ 0.65 cm^2^/m^2^ and ≤ 0.85 cm^2^/m^2^, and did not occur if EOAI > 0.85 cm^2^/m^2^.

### 2.5. Definition of Endpoints

The first object was relative LVMI regression to assess the degree of reverse remodeling of LV hypertrophy and the extent of LVMI regression. The second endpoint was MACCE, i.e., death due to a cardiac cause, myocardial infarction, all types of strokes, and any reintervention and rehospitalization due to heart failure [27,28].

### 2.6. Data Analysis

The differences in echocardiography parameters (including geometry and hemodynamic function) at pre-operation and follow-up were compared using paired sample *t*-tests. The association between echocardiography parameters of interest at follow-up and relative LVMI regression was tested using linear regression analysis. The association between echocardiography parameters of interest at follow-up and the risk of subsequent MACCE was assessed using the Cox proportional hazards model. In addition, the association between changes in the selected echocardiography parameters (EOAI, MPG, Z_VA_, and SWL) from pre-operation to follow-up and relative LVMI regression was evaluated using linear regression analysis. The association between changes in selected echocardiography parameters and the risk of MACCE was investigated using the Cox proportional hazards model. Due to the small sample size (*n* = 120) and the limited number of events (44 patients with MACCE), the above regression models were adjusted for age, sex, hypertension, systolic dysfunction (preoperative LVEF < 50%), and dilated LVEDD (>55 mm). Finally, the optimal cutoff point of relative LVMI regression to discriminate subsequent MACCE was determined using maximally selected rank statistics. All tests were two-tailed and the level of statistical significance was set at *p* < 0.05. The analysis to determine the optimal cutoff point was performed using the “survminer” package (version: 0.4.9) in R 4.2.0. The other statistical analyses were performed using SPSS 26 (IBM SPSS Inc., Chicago, IL, USA).

## 3. Results

### 3.1. Patients’ Characteristics

One hundred and twenty patients underwent echocardiography at pre-operation and follow-up 6–24 months after the operation. Their mean age was 67.7 years (standard deviation [SD] = 10.2 years) and 55% of the patients were male. Only eight patients (6.7%) had received previous AVR surgery. Approximately two-thirds 82 (68.3%) of the patients received mini-invasive AVR surgery. The most prevalent comorbidity was hypertension 91 (75.8%), followed by hyperlipidemia 79 (65.8%), coronary artery disease 51 (42.5%), and diabetes 48 (40%). Small annulus size <21 mm 71 (59.2%). At pre-operation, 22 patients (18.3%) showed systolic dysfunction and 26 (21.7%) had dilated LVEDD. Most of the patients (*n* = 117, 97.5%) received a bioprosthesis; 48 of the prostheses were porcine and 44 bovine, while 25 were sutureless valves (Table 1). In three cases, mechanical valves were implanted (1 St. Jude Standard size-21 case, 1 St. Jude Regent size-21 case, 1 St. Jude Regent size-23 case) and bioprosthetic valves in 117 cases, including 48 porcine valves (13 Epic cases: 2 cases in size-19, 5 cases in size-21, 5 cases in size-23, 1 case in size-25; 35 Hancock cases: 12 cases in size-21, 17 cases in size-23, 4 cases in size-25, 2 cases in size-27); 44 Bovine valves (27 Mitroflow cases: 1 case in size-19, 14 cases in size-21, 11 cases in size-23, 1 case in size-25; 9 Carpentier-Edwards Perimount cases: 5 cases in size-21, 1 case in size-23, 3 cases in size-25; 8 Carpentier-Edwards Magna cases: 1 case in size-21, 3 cases in size-23, 4 cases in size-25); and 25 Sutureless valve cases (25 Perceval cases: 8 cases in S-size, 9 cases in M-size, 5 cases in L-size, 3 cases in XL-size). Among the varying prostheses between 19 mm and 27 mm, 19 mm were used in 3 patients (2.5%), 21 mm in 39 patients (32.5%), 23 mm in 38 patients (31.7%), 25 mm in 13 patients (10.8%), and 27 mm in 2 patients (1.7%). The Appendix A provides a detailed description of the implanted valves.

### 3.2. Hemodynamic Parameters before and after the Operation

The preoperative and postoperative values of geometry and hemodynamic function are shown in Table 2. All geometric parameters, including interventricular septum thickness, LV posterior wall thickness, LV internal dimensions, and LV internal volume, decreased significantly after AVR (*p* < 0.001). All hemodynamic function parameters changed significantly after AVR, except for the RWT ratio (0.53 at pre-operation vs. 0.54 at follow-up; *p* = 0.704).

### 3.3. Association between Follow-Up Hemodynamic Parameters and Relative LVMI Regression

The results of the analysis of the association between hemodynamic parameters at follow-up and relative LVMI regression are shown in Table 3. They show that the higher the MPG, higher SWL, the lower the EOAI value, the presence of PPM, higher postoperative LVMI, and concentric or eccentric LV remodeling mode were significantly associated with lower relative LVMI regression (*p* < 0.05). In addition, preoperative LVMI was significantly negatively associated with relative LVMI regression during follow-up (regression coefficient [*B*], −0.22; 95% confidence interval [CI], −0.30 to −0.13). The relationships between preoperative LVMI and postoperative LVMI and relative LVMI regression are illustrated in Figure 1A,B.

### 3.4. Association between Follow-Up Hemodynamic Parameters and the Risk of MACCE

Over the mean follow-up period of 3.6 years (SD = 2.0 years), 19 patients suffered from stroke, 31 patients were hospitalized for heart failure, 7 patients underwent a redo-AVR operation, and 13 patients died. This means that 44 patients (36.7%) experienced a MACCE. The results of the analysis of the association between hemodynamic parameters at follow-up and the risk of MACCE are shown in Table 4. Z_VA_ (hazard ratio [HR], 1.65; 95% CI 1.21–2.25), SWL (HR, 1.13; 95% CI, 1.06–1.22), the presence of PPM (HR, 2.75; 95% CI, 1.38–5.46), RWT (HR, 1.43; 95% CI, 1.14–1.79), postoperative LVMI (HR, 1.01; 95% CI, 1.004–1.02), and concentric or eccentric LV remodeling mode were significantly positively associated with the risk of MACCE. The cumulative rates of MACCE for patients with higher (>3.5 mmHg/mL/m^2^) and lower (≤3.5 mmHg/mL/m^2^) postoperative Z_VA_ are shown in Figure 2A.

### 3.5. Association between Changes in Hemodynamic Parameters and Relative LVMI Regression and Risk of MACCE

The associations between changes in hemodynamic parameters (especially EOAI, MPG, Z_VA_, and SWL) and relative LVMI regression and the risk of MACCE were analyzed. The results showed that a smaller improvement in MPG (*B*, 0.46; 95% CI, 0.26–0.67) and SWL (*B*, 0.95; 95% CI, 0.52–1.38) was significantly associated with lower relative LVMI regression (upper panel in Table 5). In addition, smaller changes in MPG (HR, 1.02; 95% CI, 1.002–1.04), Z_VA_ (HR, 1.35; 95% CI, 1.06–1.72), and SWL (HR, 1.05; 95% CI, 1.01–1.09) were significantly associated with higher risk of MACCE. Furthermore, lower relative LVM regression was significantly associated with a higher risk of subsequent MACCE (HR, 1.01; 95% CI, 1.003–1.03). The corresponding optimal cutoff of relative LVMI regression was −14%; the cumulative event rates against the cutoff are shown in Figure 2B.

## 4. Discussion

Aortic stenosis is the third most common cardiovascular disease after coronary artery disease and systemic arterial hypertension. It occurs in an estimated 25% of people >65 years old and almost 50% of people aged >85 years old in developed countries [1]. Doppler echocardiography is the method used for diagnosis of AS and surveillance after AVR. AS should be considered a disease of the left ventricle [1,29,30,31,32]. The transvalvular pressure gradient and the LV workload increase exponentially as the aortic area becomes smaller, the persistent high-pressure delayed LV mass regression [2,32,33,34]. Increased LV transmural pressure enhances coronary vascular resistance and mismatch between myocardial oxygen demand and coronary flow reserve, resulting in the development of myocardial ischemia, apoptosis of myocytes, and substituted myocardial fibrosis [35,36]. Hypertrophic remodeling adjusted to pressure overload has both adaptive and maladaptive aspects [28,37,38,39]. The adaptive response of the left ventricle to chronic pressure overload due to LV outflow obstruction is characterized by hypertrophic remodeling, resulting in LV diastolic and systolic dysfunction [2]. The transformation of LV geometry and function shows a degenerative process caused by time-dependent myocardial apoptosis and then gradually developed myocardial fibrosis. Fibrosis is a recognized marker of maladaptive LV hypertrophy and poor prognosis with further cardiac events and mortality [5,31]. The Asian is generally short in stature and has a small aortic annlus, so most were implanted with small-sized valves. We disclosed the results after AVR in the Taiwanese people.

### 4.1. Patient-Prosthesis Mismatch

Patient–prosthesis mismatch after AVR for AS is not uncommon; severe PPM occurs in 2–20% of patients and is associated with a 1.5- to 2.0-fold increase in the risk of mortality and rehospitalization for heart failure [40,41]. PPM reduces the unloading benefitinstead of increasing LV outflow afterload, and increases LV wall stress, the force over the prosthetic valve orifice, and LV inner surface area during the end systolic phase [29]. The residual LV afterload due to PPM hinders reverse remodeling and causes a deterioration process similar to that due to AS. The causes of residual hypertrophy may be divided into two categories: reversible hemodynamic causes and irreversible non-hemodynamic causes [11,39,42]. The hemodynamic component can lead to LV reverse remodeling, but the non-hemodynamic factors cause transitional sequelae from irreversible myocardial damage, leading to “pathologic” fibrotic hypertrophy with impaired LV contractility, which results in cardiomyopathy [7,39,42].

Myocardial fibrosis is an important morphological postoperative clinical outcome in severe AS and is irreversible after AVR [43]. The reduced LV mass regression after AVR is associated with inferior hemodynamic function, further cardiac events, and high readmission rates for both heart failure and repeat AVR, as well as having an impact on long-term survival [31,41,44]. PPM is also associated with faster structural valvular degeneration of bioprosthetic valves after AVR [45,46]. Most of the patients in our study (*n* = 117, 97.5%) received bioprosthetic valves, of which 48 were porcine valves, 44 were bovine valves, and 25 were sutureless valves. Because of the generally short stature of Taiwanese people, the aortic annuli were less than 21 mm in the majority of cases (71 [59.2%]) in this study. This was the reason for the severe PPM occurring in 5% and moderate PPM in 44.2% of all cases, with PPM observed in almost half of our study population (Table 1). This is a critical issue for long-term surveillance in our series.

### 4.2. Reverse Remodeling and Residual Hypertrophy

Patient-prosthesis mismatch hinders reverse remodeling and causes residual hypertrophy after AVR. Concentric hypertrophy is associated with myocardial cell apoptosis and subsequent development of fibrosis, a strong predictor of poor outcomes and mortality [30,47,48,49]. In our study, the preoperative LV geometry showed LV hypertrophy in about 113 (94.2%) patients, and the different remodeling modes revealed concentric hypertrophy in 90 (75%) patients and eccentric hypertrophy in 23 (19.2%) patients. Most remodeling modes showed RWT ratio >0.42 occupied 97 (80.8%), hypertrophy about 113 (94.2%), concentric hypertrophy 90 (75%), eccentric hypertrophy 23 (19.2%). LV dilatation, an early sign of LV function deterioration, was also noted in 26 (21.7%) (LVEDD >55 mm) of these cases. LV diastolic dysfunction was present in 118 (98.3%) cases and systolic dysfunction in 22 (18.3%) patients, meaning that long-term pressure overload induced diastolic and systolic dysfunction. This clinical finding indicates that dimensional change in the LV wall caused long-term pressure overload, leading to firstly apoptosis and then fibrosis, resulting in the transformation of LV geometry and function [43]. The progression of hemodynamic function deterioration was correlated with an alteration of LV geometry, such as LVEDD dilatation. The most frequent adaptation to AS was concentric hypertrophy, i.e., a greater RWT ratio, a physiological response to increased wall stress in order to maintain the LV ejection fraction and cardiac output [4,30,47,48,49]. However, the cost of adaptation is sub-endocardial ischemia because of the mismatch between oxygen demand and supply due to the impaired coronary reserve. The sequelae of concentric hypertrophy include reduced diastolic perfusion time, microvascular dysfunction, low coronary perfusion pressure, and genetic factors associated with LV dysfunction [5].

### 4.3. Global Left Ventricular Afterload

The global LV afterload, estimated using Z_VA_, is the sum of valvular obstruction and systemic vascular impedance. LV outflow tract obstruction elevates the transvalvular pressure gradient, which represents the valvular portion of Z_VA_. SWL can be interpreted as blood pressure normalization of the transvalvular pressure gradient [15]. SWL is based only on pressure estimates without the need to measure the flow rate. From the perspective of fluid dynamics, this index represents the energy of the work performed by the left ventricle dissipated as heat and kinetic energy because of outflow obstruction, an inverse and quadratic correlation between SWL and AVA [50]. Hypertension is a common comorbidity in AS and is associated with the negative effects of LV remodeling and increased cardiovascular morbidity and mortality [20,51,52,53]. In our series, the most prevalent comorbidity was hypertension at 91 (75.8%), followed by hyperlipidemia at 79 (65.8%), coronary artery disease 51 (42.5%), and diabetes 48 (40%) (Table 1). Hypertension and reduced systemic arterial compliance are often overlooked and hinder the beneficial effects of AVR. Hypertension causes stiffness in the systemic vasculature, increasing the arterial portion of Z_VA_. Postoperative LV hypertrophy may be improved by controlling postoperative hypertension, important for LV mass regression [51,52,53,54]. Z_VA_, measured by Doppler echocardiography, was first proposed by Briand and colleagues as an estimate of the global LV afterload [20]. Z_VA_ estimates the global LV afterload, a measure of the global load imposed on LV, as the sum of valvular obstruction and systemic vascular impedance. Z_VA_ is a risk predictor of outcome [16,18,20,55]. Reduced systemic arterial compliance due to concomitant arterial atherosclerosis and high Z_VA_ may be an index of more advanced myocardial fibrosis and dysfunction [56]. Herrmann et al. reported that Z_VA_ is highly correlated with the extent of myocardial fibrosis and systolic longitudinal shortening [56]. In summary, AS should be regarded as a systemic process involving valve stenosis and reduced systemic arterial compliance [20,55] and as a disease of the left ventricle.

### 4.4. Left Ventricular Mass Regression

The preoperative and postoperative values of LV geometry and hemodynamic function showed that all geometric parameters, including interventricular septum thickness, LV posterior wall thickness, LV internal dimension, and LV internal volume, decreased significantly after AVR. All hemodynamic function parameters also changed significantly after AVR, except for RWT ratio, which was not correlated with LVMI regression. This may be due to an irreversible non-hemodynamic cause, more fibrosis, or deterioration of preoperative function in our group. The reduced reverse remodeling of the LV mass after AVR was associated with inferior hemodynamic function, more MACCE, and also influenced long-term survival [31,41,44]. A previous study showed that the maximum LV mass regression occurs within the first six months to one year after AVR [32]; the greater LVM regression in the first year was independently associated with lower risks of all-cause death, CV death, and rehospitalization [57,58,59]. Our results are in agreement with those of previous reports [11,59,60]. They show associations between hemodynamic parameters at follow-up and relative LVMI regression (Table 3). The results showed that a lower EOAI, the presence of PPM, higher SWL, higher MPG, higher postoperative LVMI, and concentric or eccentric LV remodeling mode were significantly associated with a lower reduction or even an increment in LV mass. In addition, a higher preoperative LVMI was significantly associated with a greater reduction in LV mass. The EOAI was significantly inversely proportional to relative LV mass regression, i.e., a larger EOAI meant greater relative LVMI regression. A larger EOAI also meant that a lower MPG, less SWL, and less Z_VA_ triggered greater relative LVMI regression. Preoperative LVMI is inversely proportional to relative LVMI regression (Figure 1A) and postoperative LVMI is directly proportional to relative LVMI regression (Figure 1B). Patients with greater relative LVMI regression (an LV mass reduction larger than 14%) had better outcomes than those with lower regression (an LV mass reduction lower than 14%) (Figure 2B). In our study, PPM was not observed in the newer generation of prosthetic valves, and sutureless prosthetic valves showed a lower MPG, greater LVM reduction, and fewer MACCE. The newer-generation prostheses with a narrow suture ring and the sutureless valves without a sewing ring, with their geometry of a larger orifice area, not only offer a larger EOA but also provide better hemodynamic advantages [61,62]. In this study, Z_VA_ was not correlated with relative LVMI regression, a result different from the conclusion of Ito et al. [55]. The RWT ratio was also not correlated with relative LVMI regression, and did not show a difference between preoperative and postoperative dimensions. This may be mostly due to the non-adaptive fibrotic portion or the small number of cases in our study.

### 4.5. Major Adverse Cardiovascular and Cerebral Events

Greater relative LVMI regression was associated with fewer MACCE. Lower relative LVMI regression after AVR was correlated with inferior hemodynamic function and more MACCE. During the mean follow-up period of 3.6 ± 2 years, 19 patients suffered from stroke, 31 patients were hospitalized for heart failure, 7 patients underwent a redo-AVR operation, and 13 patients died, i.e., 44 patients (36.7%) experienced MACCE. The associations between hemodynamic parameters at follow-up and the risk of MACCE are shown in Table 4. The results showed that a higher Z_VA_ (HR, 1.65; 95% CI, 1.21–2.25), higher SWL (HR, 1.13; 95% CI, 1.06–1.22), the presence of PPM (HR, 2.75; 95% CI, 1.38–5.46), higher RWT (HR, 1.43; 95% CI, 1.14–1.79), greater postoperative LVMI (HR, 1.01; 95% CI, 1.004–1.02), and a concentric or eccentric LV remodeling mode were significantly associated with a greater risk of MACCE. The cumulative event rates of MACCE for patients with higher (>3.5 mmHg/mL/m^2^) and lower (≤3.5 mmHg/mL/m^2^) postoperative Z_VA_ are shown in Figure 2A. A postoperative Z_VA_ of >3.5 mmHg/mL/m^2^ was an independent risk predictor of clinical MACCE. Although SWL decreased after AVR, a higher persistent postoperative SWL or MPG also caused a higher number of MACCE. Z_VA_ and RWT ratio were not correlated with relative LVMI regression but had strong relationships with postoperative MACCE.

We evaluated the association between changes in hemodynamic parameters and relative LVMI regression and the risk of MACCE (Table 5). The results showed that greater changes in MPG (*B*, 0.46; 95% CI, 0.26–0.67) and SWL (*B*, 0.95; 95% CI, 0.52–1.38) were significantly associated with higher relative LVMI regression. In addition, greater changes in MPG (HR, 1.02; 95% CI, 1.002–1.04), Z_VA_ (HR, 1.35; 95% CI, 1.06–1.72), and SWL (HR, 1.05; 95% CI, 1.01–1.09) were also significantly associated with a lower risk of MACCE. Furthermore, lower relative LVMI regression was significantly associated with a higher risk of subsequent MACCE (HR, 1.01; 95% CI, 1.003–1.03). The corresponding optimal cutoff of relative LVM regression was −14%; the cumulative event rates against the cutoff are shown in Figure 2B. The higher increase in EOAI alteration revealed a greater reduction in relative LV Mass, and the extent of reduction in MPG or SWL depended on the amount of increase in EOAI. The extent of relative LVMI regression was proportional to the extent of reduction in MPG and SWL. The more reduction in MPG and SWL, the more the results showed the greater LVMI regression. However, there was no association between changes in Z_VA_ alteration and relative LVMI regression.

Higher Z_VA_ was correlated with higher MACCE, and higher relative LVMI regression was associated with fewer MACCE.

### 4.6. Strategy for Avoiding PPM and Less Reverse Remodeling

The preventive strategy is to preserve EOAI at >0.85 cm^2^/m^2^ to avoid PPM, lower the risks of MACCE, and obtain better long-term results. The EOAI, which reflects the severity of PPM, is an independent predictor of LVM regression [59]. Survival decreases with increasing severity of PPM, and the risk of readmission for heart failure increases in a stepwise fashion with increasing severity of PPM [44,54]. Persistence of residual LV afterload after AVR (such as persistent higher blood pressure or residual transvalvular gradient) is concomitant with lower LV mass regression and impaired recovery of LV function [10,34,58]. Greater enlargement of aortic EOA and relief of pressure overload are stimuli for LV reverse remodeling. The purpose of AVR is to lower the global LV afterload, lessen LV wall stress, and reduce the LV workload, thus resulting in greater LV mass regression and promotion of reverse remodeling. To avoid PPM, the EOAI should be determined after annulus sizing in advance of the operation; then, a prosthetic valve, such as a newer-generation or sutureless valve or a mechanical valve, should be chosen according to the projected EOA reference table from a previous report [40]. In our study, the extent of LVMI regression was strongly correlated with subsequent MACCE. Z_VA_ can be calculated using the projected EOA following AVR, but the extent of LVMI regression is obtained using echocardiography a few months after the operation. Z_VA_ may be used to predict prognosis in advance and earlier than prediction with the extent of LV mass regression.

### 4.7. Clinical Implications

The Asian is generally short in stature and has a small aortic annulus, so most small-sized valves were implanted. We revealed the results of post-aortic valve replacement, patient-prosthesis mismatch after AVR, is not an uncommon occurrence. PPM affects LVMI regression; it is an obstacle to reverse remodeling and leads to residual hypertrophy. We found preoperative LVMI to be a predictor of LVMI regression; hemodynamic parameters, including EOAI, MPG, SWL, LVMI, and remodeling mode, were also predictors too. We also found that the risk predictors of MACCE included the hemodynamic parameters Z_VA_, SWL, EOAI, MPG, LVMI regression, and remodeling mode. Hypertension was the common comorbidity in AS; therefore, postoperative blood pressure control is important to preserve the benefits of AVR. The preventive strategy should involve choosing newer-generation prostheses, supra-annular implantation, or aortic root augmentation. Preventive strategy is a critical issue for long-term surveillance in Taiwanese people.

### 4.8. Limitations

Firstly, this study was a nonrandomized retrospective investigation with a small number of patients. Secondly, nearly half of the prostheses were 19 and 21 mm for the relatively small annulus in the population studied. Thirdly, the quality of echocardiographic assessment is observer dependent. Finally, the surveillance interval of the echocardiographic data of the 120 patients varied. LV mass regression is a complex phenomenon that is influenced by patient-related and prosthesis-related factors. Residual hypertrophy after an unload of LV pressure overload may be due to irreversible changes in interstitial fibrosis owing to long-term disease, and could affect the results.

## 5. Conclusions

This study confirmed that the relationship between PPM and increased cardiac events can be observed at any degree of PPM. Variation in LVMI regression was observed after AVR and was predicted by EOAI, MPG, SWL, remodeling mode, and preoperative and postoperative LVMI. Risk predictors for postoperative MACCE were EOAI, MPG, SWL, RWT, Z_VA_, and remodeling mode. The reduction in global LV afterload following AVR has an important role in decreasing LV mass and enhancing LV reverse remodeling. Values of Z_VA_ >3.5 mmHg/mL/m^2^ predict MACCE earlier than LVMI regression. Attention should be paid to providing adequate blood pressure control with antihypertensive agents after AVR. Strategies to prevent PPM include preoperative planning with projected EOA reference tables, aggressive valve sizing, adjunctive intraoperative procedures (aortic root augmentation), and using newer-generation valve designs (e.g., supra-annular positioning with thinner sewing rings), new-generation bi-leaflet mechanical valves, and sutureless bioprosthetic valves. Earlier valve replacement should be scheduled before irreversible maladaptive LV hypertrophy and predictors of myocardial fibrosis should be identified with echocardiography.

## Figures and Tables

**Figure 1 jcm-11-07482-f001:**
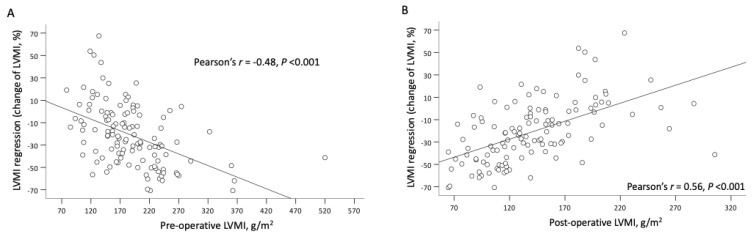
Preoperative LVMI is inversely proportional to relative LVMI regression (**A**) and postoperative LVMI is directly proportional to relative LVMI regression (**B**).

**Figure 2 jcm-11-07482-f002:**
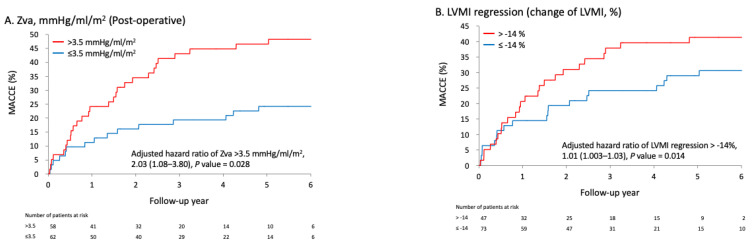
The cumulative event rates of MACCE for patients with higher (>3.5 mmHg/mL/m^2^) and lower (≤3.5 mmHg/mL/m^2^) postoperative Z_VA_ are shown in (**A**). The corresponding optimal cutoff of relative LVM regression was −14%; the cumulative event rates against the cutoff are shown in (**B**).

**Table 1 jcm-11-07482-t001:** BaselineCharacteristics, comorbidities, and valve-related characteristics of 120 study patients according to patient-prosthesis mismatch.

		Patient-Prosthesis Mismatch		
Variable	Total (*n* = 120)	Moderate/Severe (*n* = 59)	None (*n* = 61)	*p*
Baseline characteristics				
Age, year	67.7 ± 10.2	68.1 ± 11.4	67.4 ± 8.9	0.685
Body mass index, kg/m^2^	25.6 ± 4.7	25.7 ± 5.3	25.4 ± 4.0	0.720
Male sex	66 (55.0)	32 (54.2)	34 (55.7)	0.869
Body surface area, m^2^	1.66 ± 0.17	1.67 ± 0.16	1.65 ± 0.19	0.506
Smoking	26 (21.7)	14 (23.7)	12 (19.7)	0.590
Surgery-related variables				
Previous AVR surgery	8 (6.7)	4 (6.8)	4 (6.6)	0.961
Concomitant CABG	23 (19.2)	15 (25.4)	8 (13.1)	0.087
Mini-invasive surgery	82 (68.3)	36(30)	46(38.3)	0.090
Comorbidities				
Coronary artery disease	51 (42.5)	27 (45.8)	24 (39.3)	0.477
Hypertension	91 (75.8)	47 (79.7)	44 (72.1)	0.335
Diabetes	48 (40.0)	26 (44.1)	22 (36.1)	0.371
Hyperlipidemia	79 (65.8)	37 (62.7)	42 (68.9)	0.478
Stroke	18 (15.0)	11 (18.6)	7 (11.5)	0.272
Chronic kidney disease (including dialysis)	33 (27.5)	18 (30.5)	15 (24.6)	0.468
Valve-related features				
Bicuspid	46 (38.3)	18 (30.5)	28 (45.9)	0.083
Annulus size <21 mm	71 (59.2)	41 (69.5)	30 (49.2)	0.024
Ascending aortic >30 mm	69 (57.5)	34 (57.6)	35 (57.4)	0.978
Pre-operation left ventricular status				
LVEF <50 %	22 (18.3)	11 (18.6)	11 (18.0)	0.931
LVESD >40 mm	24 (20.2)	10 (17.2)	14 (23.0)	0.438
LVEDD >55 mm	26 (21.7)	12 (20.3)	14 (23.0)	0.728
Remodel mode				0.479
Normal	0 (0.0)	0 (0.0)	0 (0.0)	
Physiologic	7 (5.8)	5 (8.5)	2 (3.3)	
Eccentric	23 (19.2)	11 (18.6)	12 (19.7)	
Concentric	90 (75.0)	43 (72.9)	47 (77.0)	
Valve type				0.244
Mechanical	3 (2.5)	0 (0.0)	3 (4.9)	
Bioprosthesis	117 (97.5)	59 (100.0)	58 (95.1)	
Porcine	48 (41.0)	36 (61.0)	12 (20.7)	<0.001
Bovine	44 (37.6)	23 (39.0)	21 (36.2)	0.757
Sutureless	25 (21.4)	0 (0.0)	25 (43.1)	<0.001
Patient-prosthesis mismatch				<0.001
None	61 (50.8)	0 (0.0)	61 (100.0)	
Moderate	53 (44.2)	53 (89.8)	0 (0.0)	
Severe	6 (5.0)	6 (10.2)	0 (0.0)	
Follow-up year	3.6 ± 2.0	4.1 ± 2.2	3.1 ± 1.6	0.004

Abbreviations: LVEF is the left ventricular ejection fraction; LVESD is the left ventricular end systolic diameter; LVEDD is the left ventricular end diastolic diameter. Data were expressed as frequency (percentage) or mean ± standard deviation.

**Table 2 jcm-11-07482-t002:** Comparison of hemodynamic parameters at pre-operation and follow up.

Hemodynamic Parameter	Pre-Operation	Follow Up	*p* Value
Dimension (geometry)			
IVST, mm	13.9 ± 3.0	12.5 ± 2.8	<0.001
PWT, mm	13.1 ± 2.4	11.9 ± 2.4	<0.001
LVEDD, mm	50.5 ± 7.5	45.4 ± 6.6	<0.001
LVESD, mm	33.1 ± 9.4	28.0 ± 6.7	<0.001
LVEDV, mL	123.7 ± 43.7	96.9 ± 34.4	<0.001
LVESV, mL	50.1 ± 36.6	32.5 ± 22.2	<0.001
Hemodynamic function			
LVEF, %	62.9 ± 16.2	67.4 ± 12.6	0.003
Z_VA_, mmHg/mL/m^2^	4.4 ± 1.9	3.5 ± 1.0	<0.001
Stroke work loss, %	0.23 ± 0.09	0.09 ± 0.04	<0.001
EOAI, cm^2^/m^2^	0.49 ± 0.19	0.96 ± 0.31	<0.001
Mean PG, mmHg	41.9 ± 19.0	13.5 ± 7.0	<0.001
RWT, %	0.53 ± 0.14	0.54 ± 0.13	0.704
LVMI, g/m^2^	188.0 ± 64.0	140.2 ± 47.9	<0.001

Abbreviation: IVST, interventricular septum thickness; PWT, posterior wall thickness; LVEDD, left ventricular end-diastolic diameter; LVESD, left ventricular end-systolic diameter; LVEVD, left ventricular end-diastolic volume; LVEVS, left ventricular end-systolic volume; LVEF, left ventricular ejection fraction; Z_VA_, valvulo-arterial impedance; EOAI, effective orifice area index; PG, pressure gradient; RWT, relative wall thickness; LVMI, left ventricular mass index. Data were presented as mean ± standard deviation.

**Table 3 jcm-11-07482-t003:** Association of interested hemodynamic parameters at follow up with relative LVMI regression (change of LVMI, expressed as percentage).

Hemodynamic	Unadjusted Analysis		Adjusted Analysis *	
Parameter	*B* (95% CI)	*p*	*B* (95% CI)	*p*
Pre-operative LVMI, g/m^2^	−0.21 (−0.29, −0.12)	<0.001	−0.22 (−0.30, −0.13)	<0.001
LVEF, %	0.02 (−0.38, 0.41)	0.924	−0.001 (−0.48, 0.48)	0.998
Z_VA_, mmHg/mL/m^2^	1.15 (−3.80, 6.10)	0.648	0.99 (−3.86, 5.84)	0.688
Z_VA_, mmHg/mL/m^2^				
≤3.5	Reference		Reference	
>3.5	1.83 (−8.05, 11.71)	0.716	1.99 (−7.63, 11.61)	0.685
Stroke work loss, %	0.89 (−0.08, 1.87)	0.072	0.97 (0.001, 1.95)	0.0497
EOAI, cm^2^/m^2^	−15.61 (−29.41, −1.80)	0.027	−16.40 (−30.08, −2.71)	0.019
EOAI, cm^2^/m^2^				
≥0.85	Reference		Reference	
<0.85	10.76 (1.06, 20.46)	0.030	10.66 (1.42, 19.89)	0.024
Mean PG, mmHg	0.66 (0.12, 1.19)	0.016	0.65 (0.12, 1.19)	0.017
Mean PG, mmHg				
<20	Reference		Reference	
≥20	6.95 (−3.10, 16.99)	0.175	6.64 (−3.04, 16.32)	0.179
RWT, per 10%	2.10 (−1.04, 5.25)	0.190	1.61 (−1.50, 4.71)	0.310
RWT				
≤0.42	Reference		Reference	
>0.42	13.23 (−0.05, 26.52)	0.051	10.55 (−3.31, 24.41)	0.136
LVMI, g/m^2^	0.32 (0.20, 0.44)	<0.001	0.34 (0.23, 0.45)	<0.001
LV remodel mode				
Normal or physiologic	Reference		Reference	
Concentric or eccentric	27.02 (16.87, 37.18)	<0.001	26.79 (17.26, 36.32)	<0.001

Abbreviation: LVM, left ventricular mass; LVMI, left ventricular mass index; *B*, regression coefficient; CI, confidence interval; LVEF, left ventricular ejection fraction; Z_VA_, valvulo-arterial impedance; EOAI, effective orifice area index; PG, pressure gradient; RWT, relative wall thickness; LV, left ventricular; LVEDD, left ventricular end-diastolic diameter; * Adjusted for age, sex, hypertension, LVEF < 50 and LVEDD > 55.

**Table 4 jcm-11-07482-t004:** Association of interested hemodynamic parameters at follow up with the risk of MACCE.

Hemodynamic	Unadjusted Analysis		Adjusted Analysis *	
Parameter	HR (95% CI)	*p*	HR (95% CI)	*p*
Pre-operative LVMI, g/m^2^	1.002 (0.998–1.007)	0.284	1.00 (0.995–1.005)	0.929
LVEF, %	0.99 (0.97–1.01)	0.486	1.00 (0.98–1.03)	0.823
Z_VA_, mmHg/mL/m^2^	1.59 (1.23–2.06)	<0.001	1.65 (1.21–2.25)	0.002
Z_VA_, mmHg/mL/m^2^				
≤3.5	Reference		Reference	
>3.5	2.28 (1.22–4.26)	0.010	2.03 (1.08–3.80)	0.028
Stroke work loss, %	1.11 (1.03–1.20)	0.005	1.13 (1.06–1.22)	0.001
EOAI, cm^2^/m^2^	0.08 (0.02–0.38)	0.001	0.06 (0.01–0.35)	0.001
EOAI, cm^2^/m^2^				
≥0.85	Reference		Reference	
<0.85	2.62 (1.35–5.11)	0.005	2.75 (1.38–5.46)	0.004
Mean PG, mmHg	1.06 (1.02–1.10)	0.001	1.07 (1.03–1.11)	<0.001
Mean PG, mmHg				
<20	Reference		Reference	
≥20	2.21 (1.17–4.20)	0.015	2.43 (1.26–4.71)	0.008
RWT, per 10%	1.36 (1.10–1.67)	0.004	1.43 (1.14–1.79)	0.002
RWT				
≤0.42	Reference		Reference	
>0.42	1.34 (0.53–3.39)	0.543	1.07 (0.42–2.74)	0.885
LVMI, g/m^2^	1.01 (1.004–1.01)	0.001	1.01 (1.004–1.02)	0.002
LV remodel mode				
Normal or physiologic	Reference		Reference	
Concentric or eccentric	2.72 (1.07–6.92)	0.036	3.32 (1.24–8.89)	0.017

Abbreviations: MACCE, major adverse cardiac and cerebrovascular events; HR, hazard ratio; CI, confidence interval; LVEF, left ventricular ejection fraction; Z_VA_, valvulo-arterial impedance; EOAI, effective orifice area index; PG, pressure gradient; RWT, relative wall thickness; LVMI, left ventricular mass index; LV, left ventricular. * Adjusted for age, sex, hypertension, LVEF < 50 and LVEDD > 55.

**Table 5 jcm-11-07482-t005:** Association of change of hemodynamic parameters with relative LVMI regression (change of LVMI, expressed as percentage) and the risk of MACCE.

	Unadjusted Analysis		Adjusted Analysis *	
Outcome/ Parameters	*B* or HR (95% CI)	*p*	*B* or HR (95% CI)	*p*
LVM regression				
EOAI, cm^2^/m^2^	−10.57 (−24.76, 3.62)	0.144	−12.74 (−25.57, 0.10)	0.052
Mean PG, mmHg	0.44 (0.23, 0.65)	<0.001	0.46 (0.26, 0.67)	<0.001
Z_VA_, mmHg/mL/m^2^	−0.75 (−3.36, 1.85)	0.572	1.06 (−1.53, 3.65)	0.423
Stroke work loss, %	0.92 (0.48, 1.36)	<0.001	0.95 (0.52, 1.38)	<0.001
MACCE				
EOAI, cm^2^/m^2^	0.19 (0.08–0.44)	<0.001	0.27 (0.11–0.66)	0.004
Mean PG, mmHg	1.02 (1.003–1.04)	0.021	1.02 (1.002–1.04)	0.031
Z_VA_, mmHg/mL/m^2^	1.41 (1.15–1.73)	0.001	1.35 (1.06–1.72)	0.015
Stroke work loss, %	1.05 (1.02–1.09)	0.005	1.05 (1.01–1.09)	0.011
LVMI regression (change of LVMI, %)	1.008 (0.998–1.018)	0.127	1.01 (1.003–1.03)	0.014

Abbreviations: LVM, left ventricular mass; LVMI, left ventricular mass index; EOAI, effective orifice area index; MACCE, major adverse cardiac and cerebrovascular events; *B*, regression coefficient; HR, hazard ratio; CI, confidence interval; PG, pressure gradient. * Adjusted for age, sex, hypertension, LVEF < 50 and LVEDD > 55.

## Data Availability

The data presented in this study are available on request from the corresponding author. The data are not publicly available due to privacy and ethical reason.

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
