# Peer review of "Extent of Left Ventricular Mass Regression and Impact of Global Left Ventricular Afterload on Cardiac Events and Mortality after Aortic Valve Replacement"

_jcm, 2022, doi:10.3390/jcm11247482_

Round 1
Reviewer 1 Report
Dear authors,
first of all I like to congratulate on your nice work.
I personally always have a problem with the concept of PPM being largely dependent from patient's body weight. Recently, we simulated impact of loosing 10% weight and reduction to normal weight on the incidence of PPM at our institution, on the latter formally no more PPM exists. This reveals the conflict - should I perform more risky surgery by means of rot enlargement or should the patient try to loose some weight (which often is more difficult, but less risky).
Nonetheless, and from a scientific point of view, your paper adds some really interesting points to the common unidimensional PPM-discussion. I do not really know if that what we define as PPM really is PPM, but you showed that less LV-regression is associated with poorer outcomes. I believe in future studies we have more to evaluate on this topic and mechanisms for a better understanding of LV-remodelling.
I only have minor remarks:
- Could you outline number and sizes of the implanted valves? This would help to interpret the results a little bit better.
- I guess you have a special population of patients because of the high proportion of small sized annuli. Potentially this relates to an Asian population. Could you mention a mean BMI of your population (a high proportion of PPM in a normal weight population is more reliable than in an overweight population)
Author Response
first of all, I like to congratulate on your nice work.
I personally always have a problem with the concept of PPM being largely dependent from patient's body weight. Recently, we simulated impact of losing 10% weight and reduction to normal weight on the incidence of PPM at our institution, on the latter formally no more PPM exists. This reveals the conflict - should I perform more risky surgery by means of root enlargement or should the patient try to lose some weight (which often is more difficult, but less risky).
Nonetheless, and from a scientific point of view, your paper adds some really interesting points to the common unidimensional PPM-discussion. I do not really know if that what we define as PPM really is PPM, but you showed that less LV-regression is associated with poorer outcomes. I believe in future studies we have more to evaluate on this topic and mechanisms for a better understanding of LV-remodeling.
Author Response
Dear reviewer,
We are thankful for your important remarks for us to improve our manuscript. We have tried to answer all your questions and comments.
I only have minor remarks:
- Could you outline number and sizes of the implanted valves? This would help to interpret the results a little bit better.
Response 1:
Thank you for your comment and suggestion.
I added the detailed description about the sizes and number of the implanted valves in manuscript page 4 from line 232 to 244. We also add a supplemental table 1 to provide a detailed description of implanted valves. The contents as following:
In three cases, mechanical valves were implanted (one St. Jude Standard size-21 case, one St. Jude Regent size-21 case, one St. Jude Regent size-23 case) and bioprosthetic valves in 117 cases, including 48 porcine valves (13 Epic cases: 2 cases in size-19, 5 cases in size-21, 5 cases in size-23, 1 case in size-25; 35 Hancock cases: 12 cases in size-21, 17 cases in size-23, 4 cases in size-25, 2 cases in size-27), 44 Bovine valves (27 Mitroflow cases: 1 case in size-19, 14 cases in size-21, 11 cases in size-23, 1 case in size-25; 9 Carpentier-Edwards Perimount cases: 5 cases in size-21, one case in size-23, 3 cases in size-25; 8 Carpentier-Edwards Magna cases: one case in size-21, 3 cases in size-23, 4 cases in size-25), 25 Sutureless valve cases (25 Perceval cases: 8 cases in S-size, 9 cases in M-size, 5 cases in L-size, 3 cases in XL-size). Among the varying prostheses between 19 mm and 27 mm, 19 mm were used in 3 patients (2.5%), 21 mm in 39 patients (32.5%), 23 mm in 38 patients (31.7%), 25 mm in 13 patients (10.8%), and 27 mm in 2 patients (1.7%). The supplemental table 1 provides a detailed description of the implanted valves.
Supplemental Table 1. Detailed description of the size and brand for the implanted valves
|
Variable |
Number (Frequency %) |
|
Valve size (n=95) |
|
|
19mm |
3(2.5) |
|
21mm |
39(32.5) |
|
23mm |
38(31.7) |
|
25mm |
13(10.8) |
|
27mm |
2(1.7) |
|
Valve size for sutureless (n=25) |
|
|
S |
8(6.7) |
|
M |
9(7.5) |
|
L |
5(4.2) |
|
XL |
3(2.5) |
|
Brand |
|
|
EPIC (porcine) |
13(10.8) |
|
Hancock (porcine) |
35(29.2) |
|
Mitroflow (bovine) |
27(22.5) |
|
Perimount (bovine) |
9(7.5) |
|
Magna (bovine) |
8(6.7) |
|
Perceval (sutureless) |
25(20.8) |
|
St Jude Standard (mechanical) |
1(0.8) |
|
St Jude Regeant (mechanical) |
2(1.7) |
|
|
|
- I guess you have a special population of patients because of the high proportion of small sized annuli. Potentially this relates to an Asian population. Could you mention a mean BMI of your population (a high proportion of PPM in a normal weight population is more reliable than in an overweight population)
Response 2:
Thank you for your comment.
According to your suggestion, we modified the Table 1 for more detailed explanation about the differences between PPM group and none-PPMgroup in baseline demography, comorbidity and implanted valve types and sizes.
We divided the studied population into 3 different columns including total, PPM (moderate + severe) and none-PPM.Because of only 6 severe PPM cases, so we merged severe and moderate cases into PPM group. In addition,
we added a row for BMI (mean± SD) in comparison with total (25.6 ± 4.7), PPM (25.7 ± 5.3) and none-PPM (25.4 ± 4.0) groups. The resultsrevealed similar mean BMI and not too overweight in PPM and none-PPM group. There was no significant difference in BMI between PPM and none-PPMgroup. The detailed valve types and sizes were also described in the modified Table 1.
Table 1. Baseline characteristics, comorbidities, and valve-related features of the 120 study patients according to patient-prosthesis mismatch
|
|
|
Patient-prosthesis mismatch |
|
|
|
Variable |
Total (n = 120) |
Moderate/ severe (n = 59) |
None (n = 61) |
P |
|
Baseline characteristics |
|
|
|
|
|
Age, year |
67.7 ± 10.2 |
68.1 ± 11.4 |
67.4 ± 8.9 |
0.685 |
|
Body mass index, kg/m2 |
25.6 ± 4.7 |
25.7 ± 5.3 |
25.4 ± 4.0 |
0.720 |
|
Male sex |
66 (55.0) |
32 (54.2) |
34 (55.7) |
0.869 |
|
Body surface area, m2 |
1.66 ± 0.17 |
1.67 ± 0.16 |
1.65 ± 0.19 |
0.506 |
|
Smoking |
26 (21.7) |
14 (23.7) |
12 (19.7) |
0.590 |
|
Surgery-related variables |
|
|
|
|
|
Previous AVR surgery |
8 (6.7) |
4 (6.8) |
4 (6.6) |
0.961 |
|
Concomitant CABG |
23 (19.2) |
15 (25.4) |
8 (13.1) |
0.087 |
|
Mini-invasive surgery |
82 (68.3) |
36(30) |
46(38.3) |
0.090 |
|
Comorbidities |
|
|
|
|
|
Coronary artery disease |
51 (42.5) |
27 (45.8) |
24 (39.3) |
0.477 |
|
Hypertension |
91 (75.8) |
47 (79.7) |
44 (72.1) |
0.335 |
|
Diabetes |
48 (40.0) |
26 (44.1) |
22 (36.1) |
0.371 |
|
Hyperlipidemia |
79 (65.8) |
37 (62.7) |
42 (68.9) |
0.478 |
|
Stroke |
18 (15.0) |
11 (18.6) |
7 (11.5) |
0.272 |
|
Chronic kidney disease (including dialysis) |
33 (27.5) |
18 (30.5) |
15 (24.6) |
0.468 |
|
Valve-related features |
|
|
|
|
|
Bicuspid |
46 (38.3) |
18 (30.5) |
28 (45.9) |
0.083 |
|
Annulus size <21 mm |
71 (59.2) |
41 (69.5) |
30 (49.2) |
0.024 |
|
Ascending aortic >30 mm |
69 (57.5) |
34 (57.6) |
35 (57.4) |
0.978 |
|
Pre-operation left ventricular status |
|
|
|
|
|
LVEF <50 % |
22 (18.3) |
11 (18.6) |
11 (18.0) |
0.931 |
|
LVESD >40 mm |
24 (20.2) |
10 (17.2) |
14 (23.0) |
0.438 |
|
LVEDD >55 mm |
26 (21.7) |
12 (20.3) |
14 (23.0) |
0.728 |
|
Remodel mode |
|
|
|
0.479 |
|
Normal |
0 (0.0) |
0 (0.0) |
0 (0.0) |
|
|
Physiologic |
7 (5.8) |
5 (8.5) |
2 (3.3) |
|
|
Eccentric |
23 (19.2) |
11 (18.6) |
12 (19.7) |
|
|
Concentric |
90 (75.0) |
43 (72.9) |
47 (77.0) |
|
|
Valve type |
|
|
|
0.244 |
|
Mechanical |
3 (2.5) |
0 (0.0) |
3 (4.9) |
|
|
Bioprosthesis |
117 (97.5) |
59 (100.0) |
58 (95.1) |
|
|
Porcine |
48 (41.0) |
36 (61.0) |
12 (20.7) |
<0.001 |
|
Bovine |
44 (37.6) |
23 (39.0) |
21 (36.2) |
0.757 |
|
Sutureless |
25 (21.4) |
0 (0.0) |
25 (43.1) |
<0.001 |
|
Patient-prosthesis mismatch |
|
|
|
<0.001 |
|
None |
61 (50.8) |
0 (0.0) |
61 (100.0) |
|
|
Moderate |
53 (44.2) |
53 (89.8) |
0 (0.0) |
|
|
Severe |
6 (5.0) |
6 (10.2) |
0 (0.0) |
|
|
Follow-up year |
3.6 ± 2.0 |
4.1 ± 2.2 |
3.1 ± 1.6 |
0.004 |
Abbreviation: LVEF, left ventricular ejection fraction; LVESD, left ventricular end-systolic diameter; LVEDD, left ventricular end-diastolic diameter. Data were presented as frequency (percentage) or mean ± standard deviation.

Reviewer 2 Report
1. English language needs redaction
2. The correct expression is patient-prosthesis mismatch not prosthesis-patient mismatch
3. The definition of patient-prosthesis mismatch was not clearly given, nor the calculation method is supported by the given references. How did you obtain GOA for the prosthesis? Was it obtained by echocardiography or it was pre-calculated
4. It is not stated which prosthesis were used and their sizes. This is highly relevant when assessing PPM
5. Also it it is not clearly defined is the aim of the study to determine if the PPM is a predictive factor for lower myocardial mass regression and higher rate of MACCE or to analyze independent predictors of MACCE and lower myocardial mass regression.
6. The results are not divided according to presence of PPM and severity of PPM
7. The study did not reveal any new findings, hemodynamic effects of PPM are very well known and already published a lot
Author Response
Response to reviewer_2
We are thankful for your important remarks for us to improve our manuscript. We have tried to answer all your questions and comments.
Comments and Suggestions for Authors
- English language needs redaction
Response 1:
English was edited by NJE. I applied the service of editing English language correction and the company provide an editorial certificate.
- The correct expression is patient-prosthesis mismatch not prosthesis-patient mismatch
Response 2:
I had corrected the term prosthesis-patient mismatch and unified to patient-prosthesis mismatch in my manuscript.
- The definition of patient-prosthesis mismatch was not clearly given, nor the calculation method is supported by the given references. How did you obtain GOA for the prosthesis? Was it obtained by echocardiography, or it was pre-calculated
Response 3:
Thank you for your suggestion.
The definition of mismatch severity according to the reference 10. They were described in the manuscript from page 3, line 149 to page 4 line 198.
We defined severe PPM if EOAI<0.65cm2/m2, Moderate if EOAI 0.65 cm2/m2 and EOAI<0.85 cm2/m2, no PPM if EOAI 0.85 cm2/m2
Reference 10: Pibarot, P. and J.G. Dumesnil, Hemodynamic and clinical impact of prosthesis–patient mismatch in the aortic valve position and its prevention.Journal of the American College of Cardiology, 2000. 36(4): p. 1131-1141.
The calculation of EOA was described in the manuscript in page 3, from line 131 to line 137.
We calculated the effective orifice area by the formula continuity equation from doppler echocardiography before operation.
The preoperative aortic valve area was calculated using the continuity equation (stroke volume divided by valve flow velocity–time integral as determined by continuous-wave Doppler).
Effective orifice area (AreaLVOT VTILVOT) VTIAV.
Indexed Effective orifice area ( Effective orifice area) body surface area
We adopted the projected EOA from study of Pibarot et. al. as our postoperative EOA, then indexed to body surface area to assess the severity of PPM.
Reference 24: Pibarot, P. and J.G. Dumesnil, Prosthetic heart valves: selection of the optimal prosthesis and long-term management. Circulation, 2009. 119(7): p. 1034-48.
All these patients received baseline Doppler echocardiographic examination on 0 to 7 days before AVR then followed up between 3 to 24 months postoperatively for dimension and function parameters.
- It is not stated which prosthesis were used and their sizes. This is highly relevant when assessing PPM
Response 4:
We explained the detailed description about the types and sizes of implanted valves in the supplemental table 1. The supplementary content was in page 4 from line 232 to line 244 described as below:
In three cases, mechanical valves were implanted (one St. Jude Standard size-21 case, one St. Jude Regent size-21 case, one St. Jude Regent size-23 case) and bioprosthetic valves in 117 cases, including 48 porcine valves (13 Epic cases: 2 cases in size-19, 5 cases in size-21, 5 cases in size-23, 1 case in size-25; 35 Hancock cases: 12 cases in size-21, 17 cases in size-23, 4 cases in size-25, 2 cases in size-27), 44 Bovine valves (27 Mitroflow cases: 1 case in size-19, 14 cases in size-21, 11 cases in size-23, 1 case in size-25; 9 Carpentier-Edwards Perimount cases: 5 cases in size-21, one case in size-23, 3 cases in size-25; 8 Carpentier-Edwards Magna cases: one case in size-21, 3 cases in size-23, 4 cases in size-25), 25 Sutureless valve cases (25 Perceval cases: 8 cases in S-size, 9 cases in M-size, 5 cases in L-size, 3 cases in XL-size). Among the varying prostheses between 19 mm and 27 mm, 19 mm were used in 3 patients (2.5%), 21 mm in 39 patients (32.5%), 23 mm in 38 patients (31.7%), 25 mm in 13 patients (10.8%), and 27 mm in 2 patients (1.7%). The supplemental table 1 provides a detailed description of the implanted valves.
Supplemental Table 1. Detailed description of the size and brand for the implanted valves
|
Variable |
Number (Frequency %) |
|
Valve size (n=95) |
|
|
19mm |
3(2.5) |
|
21mm |
39(32.5) |
|
23mm |
38(31.7) |
|
25mm |
13(10.8) |
|
27mm |
2(1.7) |
|
Valve size for sutureless (n=25) |
|
|
S |
8(6.7) |
|
M |
9(7.5) |
|
L |
5(4.2) |
|
XL |
3(2.5) |
|
Brand |
|
|
EPIC (porcine) |
13(10.8) |
|
Hancock (porcine) |
35(29.2) |
|
Mitroflow (bovine) |
27(22.5) |
|
Perimount (bovine) |
9(7.5) |
|
Magna (bovine) |
8(6.7) |
|
Perceval (sutureless) |
25(20.8) |
|
St Jude Standard (mechanical) |
1(0.8) |
|
St Jude Regeant (mechanical) |
2(1.7) |
- Also, it is not clearly defined is the aim of the study to determine if the PPM is a predictive factor for lower myocardial mass regression and higher rate of MACCE or to analyze independent predictors of MACCE and lower myocardial mass regression.
Response 5:
We calculated statistical inference in various geometry and hemodynamic parameters from doppler echocardiography before and after AVR. The effect of unloaded LV afterload by AVR leads to change of LV geometry then function change. Different extent of reverse remodel depended on the adaptive and maladaptive percentage before operation. Small effective orifice area cause transvalvular energy loss, increased LV workload and reduced coronary reserve then myocardial apoptosis. PPM was one predictor of various parameters to explanation extent of LVMI regression and MACCE. Other factors like mean pressure gradient, stroke work loss was also correlated with LVMI regression. Also, ZVA, SWL, MPG, RWT were all associated with MACCE. We confirmed the follow-up EOAI<0.85, meant PPM, revealed less LVMI regression in Table 3. The more change of EOAI lead to the greater LVMI regression was observed in Table 5. Also, the PPM group also suffered from higher HR than none-PPM group in MACCE in Table 4. The larger EOAI meant lower HR in MACCE inTable 5.
Hypertension is a common comorbidity in AS and is associated with the negative effects of LV remodeling and increased cardiovascular morbidity and mortality. In our series, the most prevalent comorbidity was hypertension (75.8%). Briand and colleagues proposed the concept about the global LV afterloadand can be estimated with doppler echocardiography. The global LV afterload, valvulo-arterial impedance ZVA, is the sum of valvular obstruction and systemic vascular impedance. PPM was one of many predictors in LV mass regression and MACCE. Systemic arterial compliance explained the arterial portion of LV afterload. ZVA estimates the total LV workload and correlated with MACCE. Herrmann et al. reported that ZVA is highly associated with the extent of myocardial fibrosis and systolic longitudinal shortening, it was correlated with long-term results.
SWL can be interpreted as blood pressure normalization of the transvalvular pressure gradient.
Reverse remodel will be observed if LV afterload unloaded. EOAI, MPG and stroke work loss all were correlated with LVMI regression observed in Table 3. The greater changes of MPG and SWL, the more LVMI regression. EOAI, MPG, stroke work loss and relative wall thickness were correlated with MACCE presented in Table 4.
- The results are not divided according to presence of PPM and severity of PPM
Response 6:
Thank you for your comment.
According to your suggestion, we modified the Table 1 for further detailed explanation about the differences between PPM and none-PPM in baseline demography, comorbidity and valve types and sizes. We divided the baseline demography into 3 different columns including total, PPM (moderate +severe) and none-PPM. Because of only 6 severe PPM cases, so we merged severe and moderate PPM cases into one group. The detailed valve types and sizes were also described in the descriptive Table 1. We make another table to compare the postoperative differences between PPM and none-PPM group about geometry and function parameters.
Table 1. Baseline characteristics, comorbidities, and valve-related features of the 120 study patients according to patient-prosthesis mismatch
|
|
|
Patient-prosthesis mismatch |
|
|
|
Variable |
Total (n = 120) |
Moderate/ severe (n = 59) |
None (n = 61) |
P |
|
Baseline characteristics |
|
|
|
|
|
Age, year |
67.7 ± 10.2 |
68.1 ± 11.4 |
67.4 ± 8.9 |
0.685 |
|
Body mass index, kg/m2 |
25.6 ± 4.7 |
25.7 ± 5.3 |
25.4 ± 4.0 |
0.720 |
|
Male sex |
66 (55.0) |
32 (54.2) |
34 (55.7) |
0.869 |
|
Body surface area, m2 |
1.66 ± 0.17 |
1.67 ± 0.16 |
1.65 ± 0.19 |
0.506 |
|
Smoking |
26 (21.7) |
14 (23.7) |
12 (19.7) |
0.590 |
|
Surgery-related variables |
|
|
|
|
|
Previous AVR surgery |
8 (6.7) |
4 (6.8) |
4 (6.6) |
0.961 |
|
Concomitant CABG |
23 (19.2) |
15 (25.4) |
8 (13.1) |
0.087 |
|
Mini-invasive surgery |
82 (68.3) |
36(30) |
46(38.3) |
0.090 |
|
Comorbidities |
|
|
|
|
|
Coronary artery disease |
51 (42.5) |
27 (45.8) |
24 (39.3) |
0.477 |
|
Hypertension |
91 (75.8) |
47 (79.7) |
44 (72.1) |
0.335 |
|
Diabetes |
48 (40.0) |
26 (44.1) |
22 (36.1) |
0.371 |
|
Hyperlipidemia |
79 (65.8) |
37 (62.7) |
42 (68.9) |
0.478 |
|
Stroke |
18 (15.0) |
11 (18.6) |
7 (11.5) |
0.272 |
|
Chronic kidney disease (including dialysis) |
33 (27.5) |
18 (30.5) |
15 (24.6) |
0.468 |
|
Valve-related features |
|
|
|
|
|
Bicuspid |
46 (38.3) |
18 (30.5) |
28 (45.9) |
0.083 |
|
Annulus size <21 mm |
71 (59.2) |
41 (69.5) |
30 (49.2) |
0.024 |
|
Ascending aortic >30 mm |
69 (57.5) |
34 (57.6) |
35 (57.4) |
0.978 |
|
Pre-operation left ventricular status |
|
|
|
|
|
LVEF <50 % |
22 (18.3) |
11 (18.6) |
11 (18.0) |
0.931 |
|
LVESD >40 mm |
24 (20.2) |
10 (17.2) |
14 (23.0) |
0.438 |
|
LVEDD >55 mm |
26 (21.7) |
12 (20.3) |
14 (23.0) |
0.728 |
|
Remodel mode |
|
|
|
0.479 |
|
Normal |
0 (0.0) |
0 (0.0) |
0 (0.0) |
|
|
Physiologic |
7 (5.8) |
5 (8.5) |
2 (3.3) |
|
|
Eccentric |
23 (19.2) |
11 (18.6) |
12 (19.7) |
|
|
Concentric |
90 (75.0) |
43 (72.9) |
47 (77.0) |
|
|
Valve type |
|
|
|
0.244 |
|
Mechanical |
3 (2.5) |
0 (0.0) |
3 (4.9) |
|
|
Bioprosthesis |
117 (97.5) |
59 (100.0) |
58 (95.1) |
|
|
Porcine |
48 (41.0) |
36 (61.0) |
12 (20.7) |
<0.001 |
|
Bovine |
44 (37.6) |
23 (39.0) |
21 (36.2) |
0.757 |
|
Sutureless |
25 (21.4) |
0 (0.0) |
25 (43.1) |
<0.001 |
|
Patient-prosthesis mismatch |
|
|
|
<0.001 |
|
None |
61 (50.8) |
0 (0.0) |
61 (100.0) |
|
|
Moderate |
53 (44.2) |
53 (89.8) |
0 (0.0) |
|
|
Severe |
6 (5.0) |
6 (10.2) |
0 (0.0) |
|
|
Follow-up year |
3.6 ± 2.0 |
4.1 ± 2.2 |
3.1 ± 1.6 |
0.004 |
Abbreviation: LVEF, left ventricular ejection fraction; LVESD, left ventricular end-systolic diameter; LVEDD, left ventricular end-diastolic diameter. Data were presented as frequency (percentage) or mean ± standard deviation.
|
Geometry |
No PPM |
PPM |
P |
||||
|
IVST, mm |
12.2 |
± |
2.9 |
12.87 |
± |
2.67 |
0.22 |
|
PWT, mm |
11.5 |
± |
2.3 |
12.38 |
± |
2.42 |
0.045 |
|
LVEDd, mm |
45.0 |
± |
6.8 |
45.74 |
± |
6.52 |
0.547 |
|
LVEDs, mm |
27.4 |
± |
6.5 |
28.59 |
± |
6.86 |
0.325 |
|
LVEVs, ml |
30.9 |
± |
19.8 |
33.87 |
± |
24.36 |
0.460 |
|
LVEVd, ml |
95.8 |
± |
34.9 |
98.3 |
± |
33.59 |
0.689 |
|
RWT, (%) |
50 |
± |
10 |
55 |
± |
14 |
0.239 |
|
LV Mass, g |
226.2 |
± |
86.7 |
241.27 |
± |
83.26 |
0.332 |
|
LV Mass Index, g/m2 |
136.3 |
± |
48.2 |
144.26 |
± |
47.66 |
0.367 |
|
Absolute LVM regression, g |
-100.1 |
± |
106.7 |
-56.13 |
± |
93.39 |
0.018 |
|
Absolute LVMI regression, g |
-60.2 |
± |
65.2 |
-34.08 |
± |
56.55 |
0.019 |
|
Percentage LVMI regression, % |
-30 |
± |
30 |
-15 |
± |
28 |
0.023 |
|
Function |
|
|
|
|
|
|
|
|
Effective Orifice Area, cm2 |
2.00 |
± |
0.56 |
1.27 |
± |
0.10 |
<0.001 |
|
EOA Index, cm2/m2 |
1.20 |
± |
0.30 |
0.76 |
± |
0.06 |
<0.001 |
|
LV ejection fraction, % |
68.2 |
± |
12.0 |
66.54 |
± |
13.24 |
0.476 |
|
LV fraction shortening, % |
39.27 |
± |
8.70 |
38.13 |
± |
7.84 |
0.472 |
|
Mean Velocity, cm/s |
151.18 |
± |
36.95 |
173.07 |
± |
45.16 |
0.004 |
|
Mean Pressure gradient, mmHg |
11.71 |
± |
5.42 |
15.43 |
± |
7.89 |
0.003 |
|
Stroke work loss, % |
8 |
± |
3 |
10 |
± |
4 |
0.002 |
- The study did not reveal any new findings, hemodynamic effects of PPM are very well known and already published a lot
Response 7:
Most Taiwanese is generally short in stature and has small aortic annulus, so nearly half of studied population were implanted with small-sized valves.PPM was a critical issue in our hospital. Small EOAI is an obstacle for reverse remodeling and leads to residual hypertrophy, PPM diminishes the benefit of LV unloading effect by AVR. PPM reveals only valvular portion of global valvulo-arterial impedance. ZVA represents estimation of the global LV workload and a more suitable predictor for MACCE. In this study, a postoperative ZVA > 3.5 mmHg/ml/m2 was an independent risk predictor for MACCE. We found that statistical inference about the extent of LVMI regression lower than -14% was a cutoff value to predict MACCE. ZVA can be estimated after AVR by doppler echocardiography, but the extent of LVMI regression should be calculated by follow-up echocardiography several months later. ZVA predicts earlier than extent of LVMI regression. Both of ZVA and extent of LVMI regression are suitable risk predictors for MACCE after aortic valve replacement.

Round 2
Reviewer 2 Report
Dear authors,
After correcting the suggestions I would recommend your paper for publication.